# Measurement of Power Frequency Current including Low- and High-Order Harmonics Using a Rogowski Coil

**DOI:** 10.3390/s22114220

**Published:** 2022-06-01

**Authors:** Mohammed El-Shahat, Elsayed Tag Eldin, Nourhan A. Mohamed, Ahdab EL-Morshedy, Mohamed E. Ibrahim

**Affiliations:** 1Department of Electrical Power Engineering, Faculty of Engineering, Cairo University, Giza 12613, Egypt or elbagalaty02010@cu.edu.eg (M.E.-S.); ahdabmk@yahoo.com (A.E.-M.); 2Future University in Egypt, Cairo 11835, Egypt; 3Electromechanical Systems Institute, Housing and Building Research Center (HBRC), Giza 12611, Egypt; nourabdelrahman94@gmail.com; 4High Voltage and Dielectric Materials Lab., Faculty of Engineering, Menoufia University, Shebin El_Kom 32511, Egypt; en.ezzat@yahoo.com; 5Faculty of Technological Industry and Energy, Delta Technological University (DTU), Quwaysna 32631, Egypt

**Keywords:** current, harmonics, power frequency, reconstruction, Rogowski coil

## Abstract

The measurement of a power frequency current including low- and high-order harmonics is of great importance in calibration as well as in testing processes. Therefore, this paper presents the measurement of the power frequency current of light-emitting diode (LED) luminaires. LED luminaires were chosen as their input current includes both low- and high-order harmonics. The measurement process depends on reconstructing an LED luminaire current without using the coil parameters. Hence, the current reconstruction process is designed to be dependent on the measured characteristics of the Rogowski coil itself considering the frequency range at which the measurement process is required. An evaluation of the proposed measurement process was theoretically and experimentally carried out. A theoretical evaluation was carried out using MATALB SIMULINK software. However, the experimental evaluation was performed by building a Rogowski coil to measure the input currents of different LED luminaires having different power ratings of 300 W, 400 W, and 600 W. The currents measured using the Rogowski coil were compared with reference currents measured using a standard measurement technique. The obtained results show the efficacy of the proposed measurement method.

## 1. Introduction

Recently, the use of low-power instrument transformers or sensors with modern power systems has increased. The reason behind this spread is that these new generations of instrument transformers facilitate the processes of management and control of modern networks [1,2,3]. The Rogowski coil is one of the most common low-power instrument transformers. It was implemented for the first time in the 19th century by Walter Rogowski. However, it is extensively used in a great number of applications, especially with the advances in electronics, signal processing, and digital systems. In fact, researchers and scientists have proposed its use in many applications. Among these applications are the measurement of power frequency current [4,5,6,7,8,9,10], measurement of impulse and pulsed currents [10,11,12,13], fault detection in DC–DC converters [14,15,16,17,18,19], condition monitoring of high-voltage insulators [20,21] and power system equipment [22], smart meters [23], and many other applications. It is used to measure the power frequency current for single-core and multi-core cables and to measure impulse currents with different characteristics. In fault detection, a Rogowski coil is suggested as a transducer to predict the pollution condition of high-voltage insulators, as well as to measure the full discharges in power transformers. Also, it is used to sense power frequency currents in smart meters. The reasons behind the spread of the Rogowski coil are its multiple advantages, such as its light weight, low cost, flexibility, linearity, immunity to core saturation (its core is manufactured from non-magnetic material), and its simple design that makes it suitable for a wide range of applications. Hence, several Rogowski coil designs have been suggested in the literature. The Printed Circuit Board (PCB) design [24,25,26,27,28,29] is one of the most recent designs increasing the spread of Rogowski coil use in several electronic applications, as this design increases its suitability to be integrated with electronic systems and devices. On the other hand, Rogowski coil measurements can show some deviations from the expected values due to positional errors [30]. However, these errors in the measured results can be reduced to be lower than 0.1% by achieving an optimal coil design [30]. Thus, designing a Rogowski coil via new optimization techniques such as particle swarm can be used to achieve the desired performance [31]. Also, a Rogowski coil can be affected by external magnetic fields resulting from external currents [32]. However, this problem can be solved using shielding or two opposite coils designed to reduce this effect [32].

The power frequency current can be measured using several contactless sensors and transducers. Conventional current transformers (CTs) and Hall effect current transducers (HTs) are among the common methods for contactless power frequency current measurement [33,34,35]. Regarding these types of transducers, their cores are made of ferromagnetic materials to achieve magnetic field concentration. Therefore, linearity in the current measurement is only achieved in a determined current region. Also, core saturation can occur in both types, leading to inaccurate current measurement at high currents, which can occur under fault conditions [33]. Therefore, for a higher measured current, a core with larger dimensions should be used with these types of transducers. In contrast, a Rogowski coil has a linear response over a very wide range of measured current at smaller dimensions compared to the CT and HT due to its dependence on a non-magnetic core. In power system networks, a current transformer has a wider spread compared to a Hall effect transducer due to economic circumstances. On the other hand, the Rogowski coil differs from the traditional current transformer (CT), as the traditional CT transforms the current directly to a lower current at its secondary winding depending on its transformation ratio. However, the Rogowski coil gives a low voltage signal at the terminals of its output. This voltage signal needs to be processed to obtain the sensed current itself. The process of signal processing of the Rogowski coil terminal voltage to obtain the sensed current is called current reconstruction.

Several methods are used for current reconstruction [4,10,36,37,38]. In [4], an evaluation of digital integration was carried out. In the same research [4], digital Fourier transform (DFT) was presented to extract the fundamental component of the power frequency AC current. The differential current reconstruction technique presented in [5] is suitable only for measuring the fundamental component of the power frequency AC current that is intended to be measured. Up to now, there has been no reported use of a Rogowski coil to measure power frequency AC current including low- and high-order harmonics. Therefore, there is a need for a current reconstruction technique that has the ability to measure a power frequency AC current containing low- and high-order harmonic components. Also, the published current reconstruction techniques depend on the measured coil parameters, which can cause errors in measurements, especially with the possibility of parameter variation with temperature and frequency. So, in this paper, the measurement of a power frequency AC current including low- and high-order harmonics is presented. The measurement was carried out using a Rogowski coil due to its advantages. The current of LED luminaires was chosen as it contains low- and high-order harmonics. The measurement process is dependent on a reconstruction technique that uses the measured characteristics of the Rogowski coil itself instead of the dependence on the coil parameters, as these can be affected by frequency. The proposed measurement process was theoretically validated using MATALB SIMULINK software, and an experimental validation was carried out by measuring the currents of 300 W, 400 W, and 600 W LED luminaires.

Accordingly, this paper is organized as follows:Section 1 (Introduction): The current section that deals with the fields in which a Rogowski coil can be used, its advantages, a literature review, and the intended work in this paper;Section 2 (Proposed Measurement Process): The adopted measurement process is discussed, and the proposed current reconstruction method depending on coil characteristics is presented;Section 3 (Simulation Validation): A system is simulated using MATALB SIMULINK software to validate the adopted measurement process;Section 4 (Experimental Validation): The experimental setup is presented, and the obtained experimental results are reported and discussed;Section 5 (Conclusions): The main conclusions from the work are extracted and summarized.

## 2. Proposed Measurement Process

Consider a Rogowski coil fitted around a conductor as shown in Figure 1, and suppose that the current passing through the conductor contains low- and high-order harmonics. Therefore, the mathematical expression of the current can be written as: (1)$$i\left(t\right)=I_{m}\sin\left(\omega t\pm \theta \right)+\sum_{2}^{n}I_{mn}\sin\left(n\omega t\pm \varphi_{n}\right)$$
where *I_m_* is the maximum current of the fundamental component (A), *ω* is the angular frequency of the fundamental component (rad/s), *θ* is the phase angle of the fundamental component (rad), *I_mn_* is the maximum current of the *n*-th harmonic (A), *t* is the time in seconds, and *φ_n_* is the phase angle of the *n*-th harmonic (rad).

As the Rogowski coil is fitted around the conductor carrying the pre-described current, there are induced EMFs in the coil for the fundamental component and each component of the harmonic currents. The induced EMF can be computed from Faraday’s law as follows:(2)$$\;\;e=M\frac{di}{dt}$$
where *e* is the induced EMF in the coil (V), and *M* is the mutual inductance (H).

Substituting (1) into (2), the induced EMF can be computed from: (3)$$e\left(t\right)=M\omega [I_{m}\cos\left(\omega t\pm \theta \right)+\sum_{2}^{n}nI_{mn}\cos\left(n\omega t\pm \varphi_{n}\right)]$$

In fact, the output voltage at the coil terminals differs from the induced EMF in magnitude and phase angle, especially at higher frequencies. This is due to the effect of coil self-inductance (*L_c_*), coil resistance (*R_c_*), and coil stray capacitance (*C_o_*). Hence, the equivalent circuit of the coil, as reported in many studies in the literature [1,2,3,4], represents the coil with the equivalent circuit shown in Figure 2. Thus, the output voltage at the coil terminals can be expressed as:(4)$$v_{o}\left(t\right)=V_{om}\sin\left(\omega t\pm \beta \right)+\sum_{2}^{n}V_{omn}\sin\left(n\omega t\pm \beta_{n}\right)]$$
where *v_o_* is the output voltage at the Rogowski coil terminals (V), *V_om_* is the maximum output voltage of the fundamental component (V), *β* is the phase angle of the fundamental component of the output voltage (rad), *V_omn_* is the maximum output voltage of the *n*-th harmonic (V), and *β_n_* is the phase angle of the *n*-th harmonic (rad). 

To reconstruct the current correctly, especially at higher frequencies, the output voltage should not be taken instead of the induced EMF. Taking the output voltage instead of the induced EMF can be accepted only for low-frequency currents due to the smaller difference between them as a result of the lower effect of the Rogowski coil circuit parameters. Considering high-frequency currents, the errors in magnitude and phase angles between the output voltage and the induced EMF will result in severe errors in the reconstructed current. Therefore, the reconstruction method in this paper is designed to depend on the measured characteristics of the coil. The measured characteristics of the coil can be obtained by injecting 1 A current at different frequencies ranging from the fundamental to the desired value of the high-order harmonic. This process should be carried out using a calibrated source. At each frequency, the maximum value of the output voltage and the phase difference between the injected current and the output voltage waveforms (*θ_e_*) are recorded. So, there are two relations can be captured from this process. The first relation is between the maximum output voltages at the Rogowski coil terminals and the frequency. However, the second relation is the difference in phase angles between the injected current and the output voltage at the Rogowski coil terminals versus the frequency. In our opinion, this method gives an accurate representation of the coil at low and high frequencies as it does not depend on coil parameters that may be affected by skin and leakage effects, especially at higher frequencies. Also, this method is easy to carry out as the coil characteristics are measured only once after manufacturing for every model of the manufactured coils.

To clearly demonstrate the adopted current reconstruction technique, consider a distorted current passing through the conductor in Figure 1. Also, consider this current having an n-th harmonic current with a value of *I_mn_* and a phase angle of *φ_n_*. This component induces an EMF in the coil and a voltage at its terminals. The voltage at the coil terminals will have a maximum value of *V_omn_* with a phase angle of *β_n_*. If the output voltage *V_omn_* is divided by the maximum recorded voltage of the measured characteristics (the input voltage of the injected current) at the same frequency, the maximum value of this component will be identified. Also, if the phase angle error at the component frequency is added to the phase angle of the terminal voltage, then the real component angle will be obtained. Regarding the maximum current and the correct phase angle, the correct sinusoidal waveform can then be obtained.

Therefore, in this paper, the measured output voltage at the Rogowski coil terminals is analyzed using digital Fourier transform (DFT) to obtain the maximum value and phase angle of its fundamental and harmonic components up to the desired frequency. The maximum value of each component is divided by the measured maximum value at the corresponding frequency of the measured coil characteristics. Also, the phase error of the measured characteristics is added to the phase angle at the corresponding frequencies. The sinusoidal waveform of each component can be constructed and, of course, the total current can be obtained. A schematic diagram of the adopted method is shown in Figure 3.

## 3. Simulation Technique

To validate the proposed measurement process theoretically, MATLAB SIMULINK was used. The Rogowski coil was simulated according to the equivalent circuit shown previously in Figure 2 and by using the equations:(5)$$e\left(t\right)=M\frac{di}{dt}$$
and
(6)$$e\left(t\right)=R_{c}i_{s}+L_{c}\frac{di_{s}}{dt}+\frac{1}{C_{o}}\int i_{s}dt$$
where *i_s_* is the current passing through the Rogowski coil winding (A).

The two Equations (5) and (6) were modeled and simulated as shown schematically in Figure 4. A current consisting of a fundamental component and only two harmonic components for simplicity was simulated as a reference current. The fundamental component of the simulated current was simulated using a sinusoidal waveform with a peak value of 10 A, a frequency of 50 Hz, and a phase angle of 0°. The two harmonic components were chosen to have frequencies of 150 Hz and 200 Hz to simulate odd and even harmonic orders. The third harmonic component (150 Hz) was simulated using a sinusoidal waveform with a peak value of 20% of the fundamental component peak and a phase angle of 0°. The fourth harmonic component (200 Hz) was simulated with a sinusoidal waveform with a peak value of 30% of the fundamental component peak, and its phase angle was 0°.

The simulated Rogowski coil was chosen to have winding resistance of 2.4 Ω, winding inductance of 1.2 mH, a stray capacitance of 10.3 nF, and a mutual inductance of 1.8 µH. Considering the simulated coil, a sinusoidal current of 1 A peak value with frequencies ranging from 30 Hz to 100 kHz was injected through the primary winding (conductor). The injected current was chosen to have a 0° phase angle at all frequencies. The relation between the maximum output voltages at the coil terminals and the frequency was obtained as plotted in Figure 5. Also, the relation between the difference in phase angle for the injected current and the terminal voltage was obtained as plotted in Figure 6. The maximum terminal voltages at 50 Hz, 150 Hz, and 200 Hz were obtained from Figure 5 and were used in the model to obtain the peak values of the reconstructed current components. The differences in phase angles at the same three frequencies were obtained from the relation in Figure 6 and were used to obtain the phase angles of the reconstructed current components. The adopted reconstruction technique was implemented according to the process discussed previously in Section 2 and clarified in Figure 3.

Figure 7 shows the reference current (the simulated reference current) that is intended to be measured, as well as the reconstructed current considering the pre-described method in Section 2. From this figure, accuracy in measurement was achieved. This is because the adopted current reconstruction method depends on the measured characteristics of the coil itself considering the entire range of desired frequencies. For further validation, the phase angle of the fourth harmonic of the simulated reference current was changed to −90°. The reconstruction process was then repeated. The result of this case is shown in Figure 8. Several changes in phase angles of the fundamental and third harmonics were also carried out. Their results are not presented here as the conclusions were the same. In fact, the measurement accuracy was not affected by the phase angles of the fundamental or harmonic current components. This is due to the dependency on the measured characteristics of the coil. For further validation, experiments were carried out as detailed in the next section.

## 4. Experimental Work

### 4.1. Rogowski Coil Characteristics

Firstly, an experimental Rogowski coil was built. The experimental coil was wound on a ring made from cross-linked polyethylene (XLPE). The ring had a mean diameter of 3.5 cm, and its cross-sectional area was 1 cm × 1.5 cm. The ring was wound with varnish-coated copper wire. The number of coil turns was 50. The characteristics of the coil were measured using the experimental setup shown in Figure 9. Hence, variable resistance was connected in series with a variable-frequency AC source to adjust the current to a 1 A peak at each frequency. The variable AC source used has the capability to inject currents up to 1.5 A with frequencies up to 5 MHz. A current transducer (100 MHz band width) was used to measure the current passing through the variable resistance. The experimental Rogowski coil was fitted to sense this current. The measured coil terminal voltage, as well as the measured current, was captured at different frequencies using a digital oscilloscope with 150 MHz band width. From this measurement, the peak values of terminal voltages were recorded within the desired frequency range, as shown in Figure 10. Also, the phase differences between the measured currents and the terminal voltages were recorded within the desired frequency range, as shown in Figure 11. The frequency range was chosen to be from 30 Hz to 35 kHz, as this range is sufficient for measuring the current of LED luminaires.

### 4.2. Experimental Validation Using the Proposed Measurement Process

To evaluate the proposed measurement process, the experimental setup shown schematically in Figure 12 was used. In this setup, an LED luminaire was connected to a 220 V, 50 Hz AC source. An LED luminaire was chosen as a load as its current contains the fundamental component in addition to low- and high-order harmonics. The current drawn by the LED luminaire was measured using a highly calibrated current transducer (CT). However, the experimental Rogowski coil was also fitted around one of the LED luminaire terminals to sense its current. A digital oscilloscope was used to capture the reference current (measured using the CT), as well as to capture the Rogowski coil terminal voltage. The measured terminal voltage was recorded as samples (25,000 samples). These samples were processed using MATLAB software (using the adopted reconstruction method) to obtain the reconstructed current. The reference and reconstructed currents were plotted together as shown below.

The experimental setup used to measure the LED luminaire current is shown in Figure 12, where three LED luminaires with 300 W, 400 W, and 600 W powers were tested. Figure 13a shows the reference and reconstructed currents for the 300 W LED luminaire. Figure 13b shows the fast Fourier transform (FFT) of the reference and reconstructed currents. Figure 14 and Figure 15 show the same results for the 400 W and 600 W luminaires, respectively. For the results shown in Figure 13, Figure 14 and Figure 15, the reconstruction process was limited to 4 kHz only. During the testing of LED luminaires according to IEC 61000-3-2 [39], the limit of harmonic orders required is up to the 40th harmonic. This means that the maximum required frequency to be measured is 2 kHz (when operating with a 50 Hz AC source). Therefore, measurement of the current with its harmonics up to 4 kHz frequency is sufficient for testing LED luminaires.

Looking to the obtained results, the reconstructed currents using the proposed measurement process accurately follow the reference currents measured using a highly calibrated CT. This is validated by the figures that show the FFT of the measured and reconstructed currents. Hence, the errors between the reference and reconstructed currents were low, as presented in Table 1. The maximum computed error was found to be 7.9%. Therefore, good accuracy in measurement considering power frequency AC currents containing low- and high-order harmonics was achieved.

For further validation, the reconstruction process was extended to 35 kHz for measuring the current of the 300 W luminaire. The obtained results considering 35 kHz in the reconstruction process are shown in Figure 16. As shown in the figure, the proposed measurement process can accurately measure power frequency AC currents with low- and high-order harmonics with only minor deviations that may be present due to the sampling effect.

## 5. Conclusions

A proposed measurement process based on the use of a Rogowski coil to measure power frequency AC currents containing low- and high-order harmonics was presented herein. The proposed measurement process was theoretically evaluated using MATLAB SIMULINK software and was also experimentally evaluated. The experimental evaluation was carried out via the measurement of LED luminaires with different power ratings. The efficacy of the proposed measurement process was thus validated. Generally, the following points can be concluded:


The adopted measurement process does not depend on the Rogowski coil parameters;Hence, the adopted method depends on using the DFT of the output voltage at the Rogowski coil terminals. Compensations are then carried out based on the measured characteristics of the coil itself to obtain a correct reconstructed current either in magnitude or in phase;The adopted method is capable of measuring power frequency AC currents containing low- and high-order harmonics with high accuracy.


## Figures and Tables

**Figure 1 sensors-22-04220-f001:**
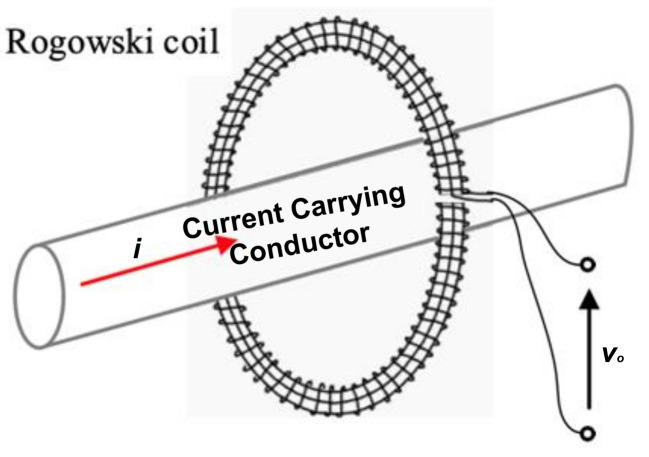
A Rogowski Coil Fitted around a Current-Carrying Conductor.

**Figure 2 sensors-22-04220-f002:**
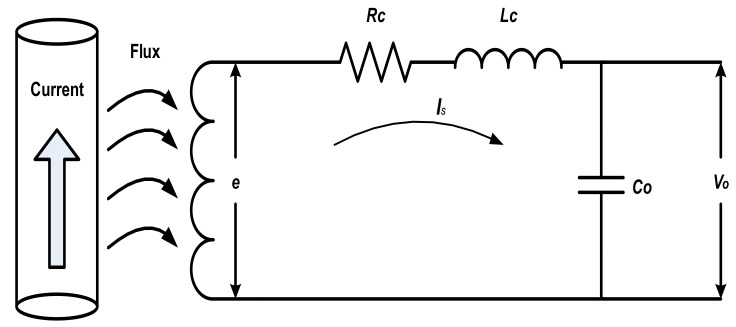
The Equivalent Circuit of a Rogowski Coil [4].

**Figure 3 sensors-22-04220-f003:**
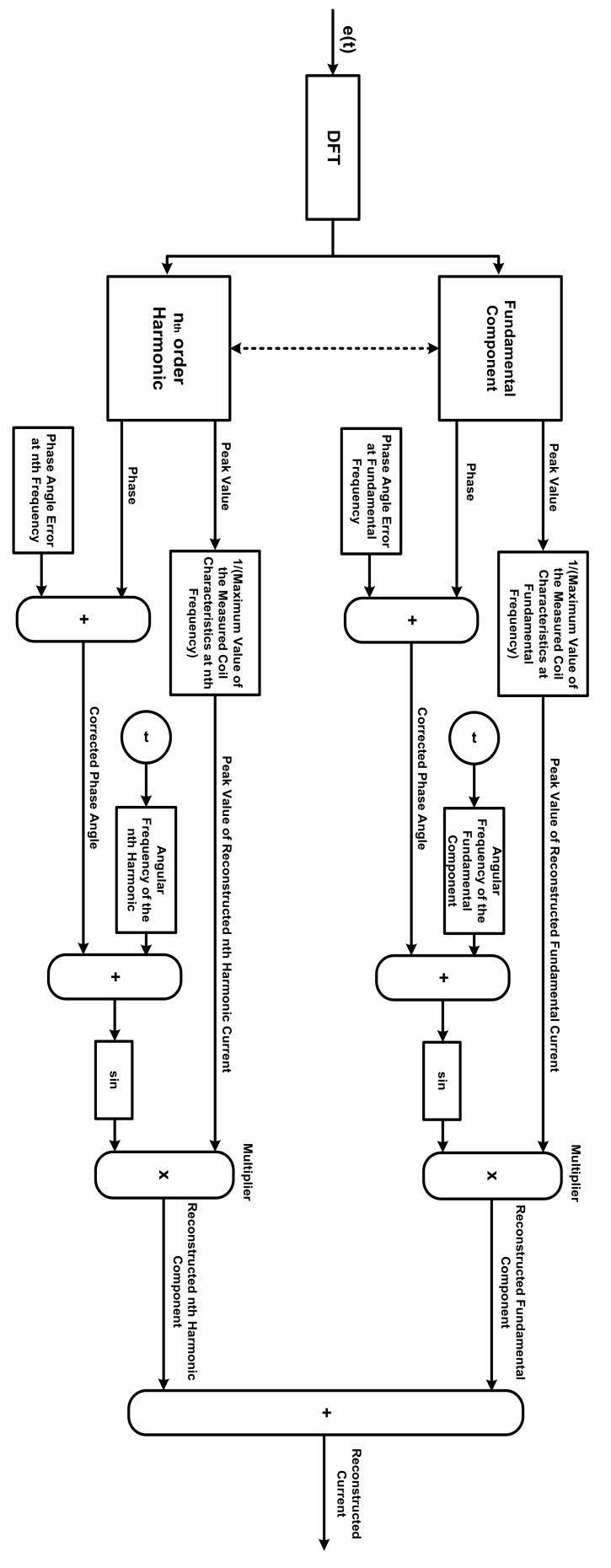
The Adopted Current Reconstruction Process.

**Figure 4 sensors-22-04220-f004:**
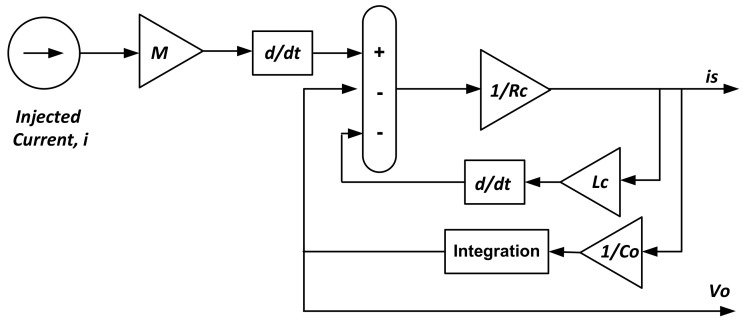
MATLAB SIMULINK Model of a Rogowski Coil.

**Figure 5 sensors-22-04220-f005:**
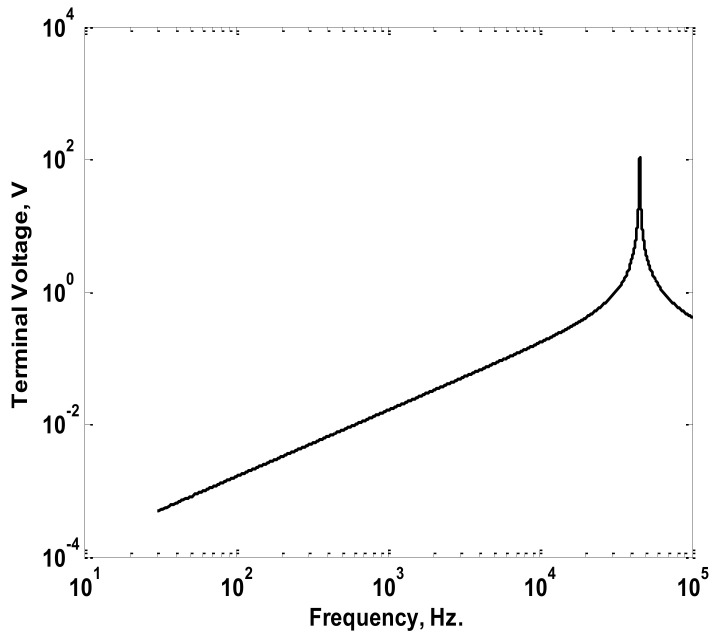
Maximum Terminal Voltages versus Frequency Characteristics of the Simulated Coil.

**Figure 6 sensors-22-04220-f006:**
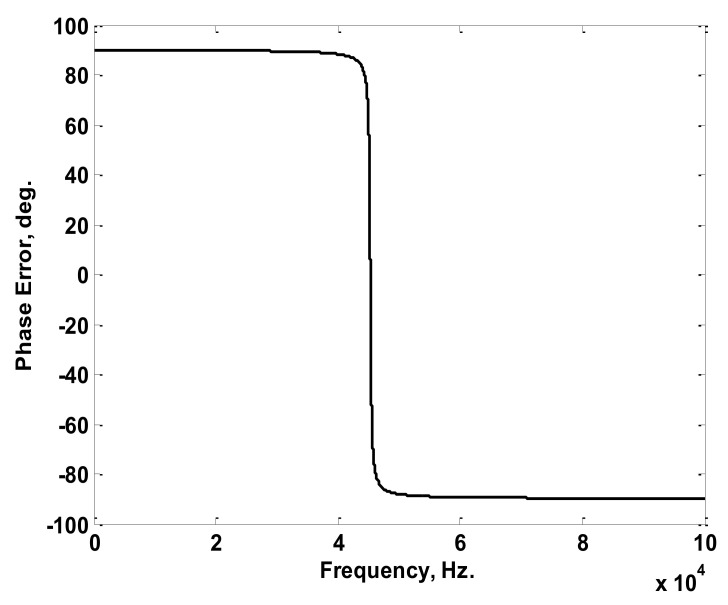
Difference in Phase Angles between the 1 A Injected Current and the Terminal Voltage versus Frequency Characteristics of the Simulated Coil.

**Figure 7 sensors-22-04220-f007:**
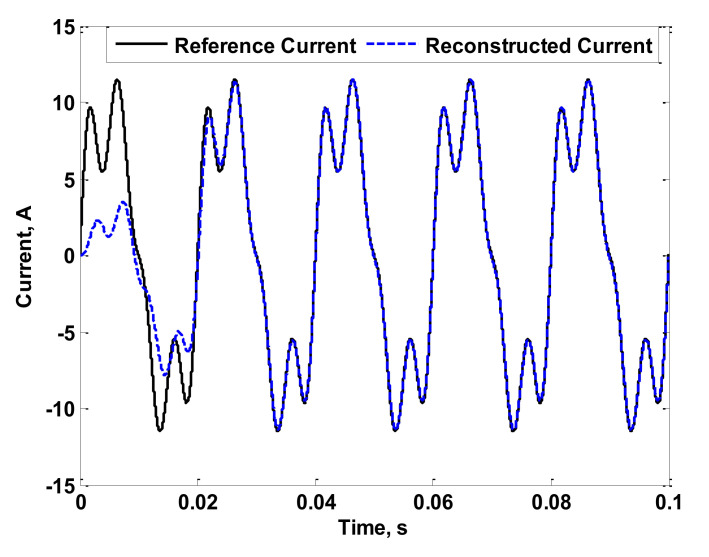
Reference and Reconstructed Currents of the Simulated Coil.

**Figure 8 sensors-22-04220-f008:**
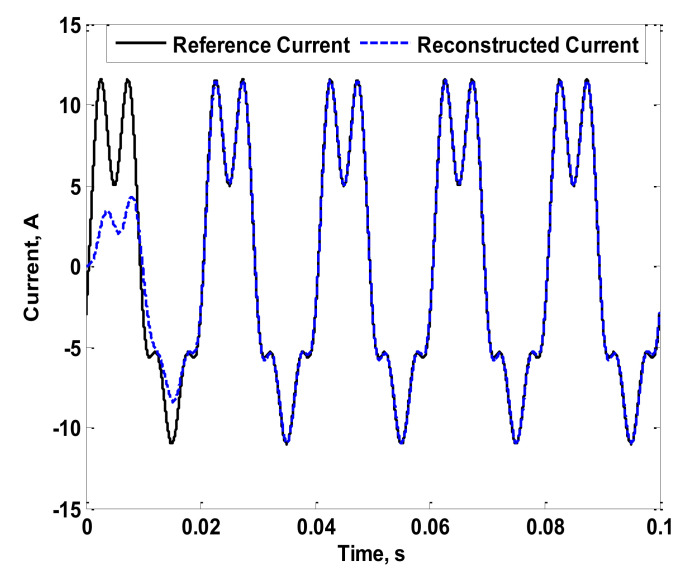
Reference and Reconstructed Currents of the Simulated Coil Considering a Change in the Phase Angle of the Fourth Harmonic.

**Figure 9 sensors-22-04220-f009:**
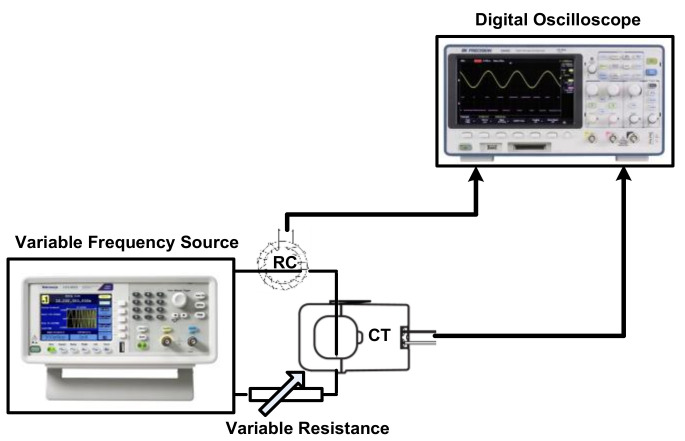
Schematic Diagram of the Experimental Setup to Measure the Rogowski Coil Characteristics.

**Figure 10 sensors-22-04220-f010:**
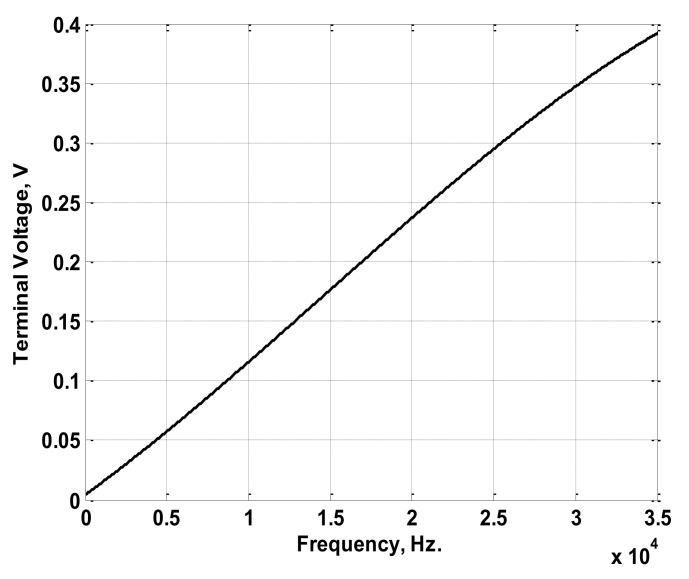
Maximum Terminal Voltages versus Frequency Characteristics of the Experimental Coil.

**Figure 11 sensors-22-04220-f011:**
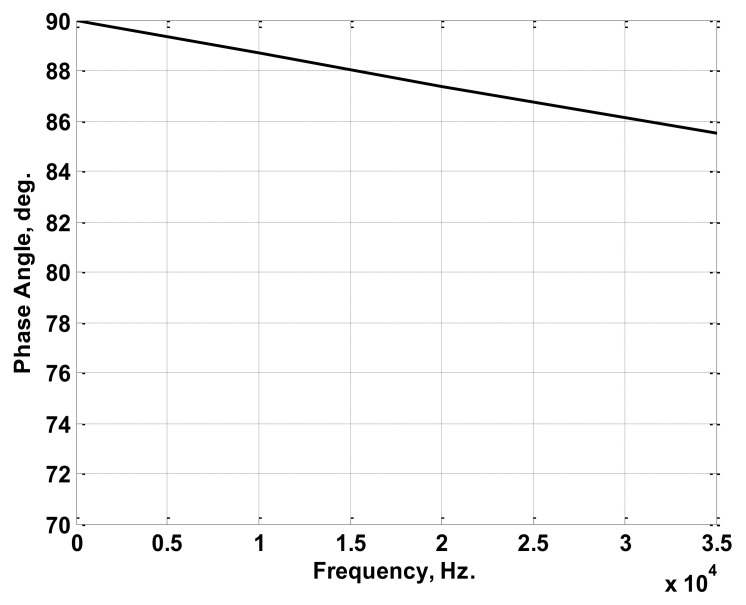
Difference in Phase Angles between the 1 A Injected Current and the Terminal Voltage versus Frequency Characteristics of the Experimental Coil.

**Figure 12 sensors-22-04220-f012:**
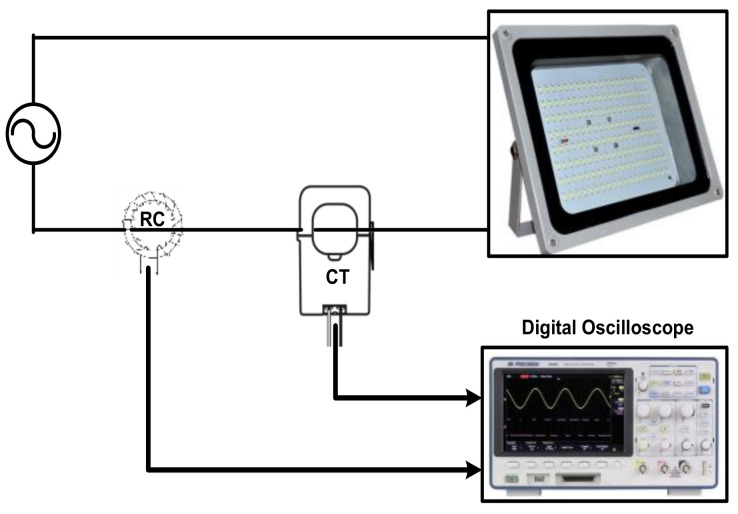
Schematic Diagram of the Experimental Setup to Measure the LED Luminaire Current.

**Figure 13 sensors-22-04220-f013:**
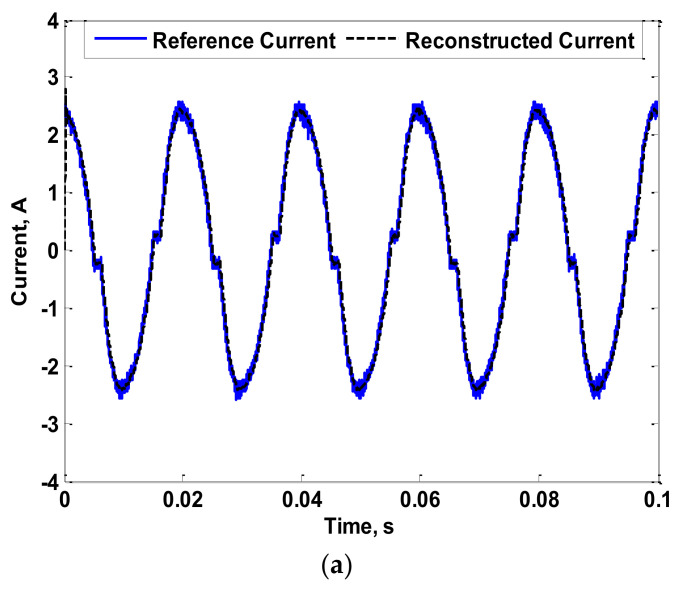
Reference and Reconstructed Currents of a 300 W luminaire: (**a**) Reference and Reconstructed Current Waveforms; (**b**) FFT of Reference and Reconstructed Currents.

**Figure 14 sensors-22-04220-f014:**
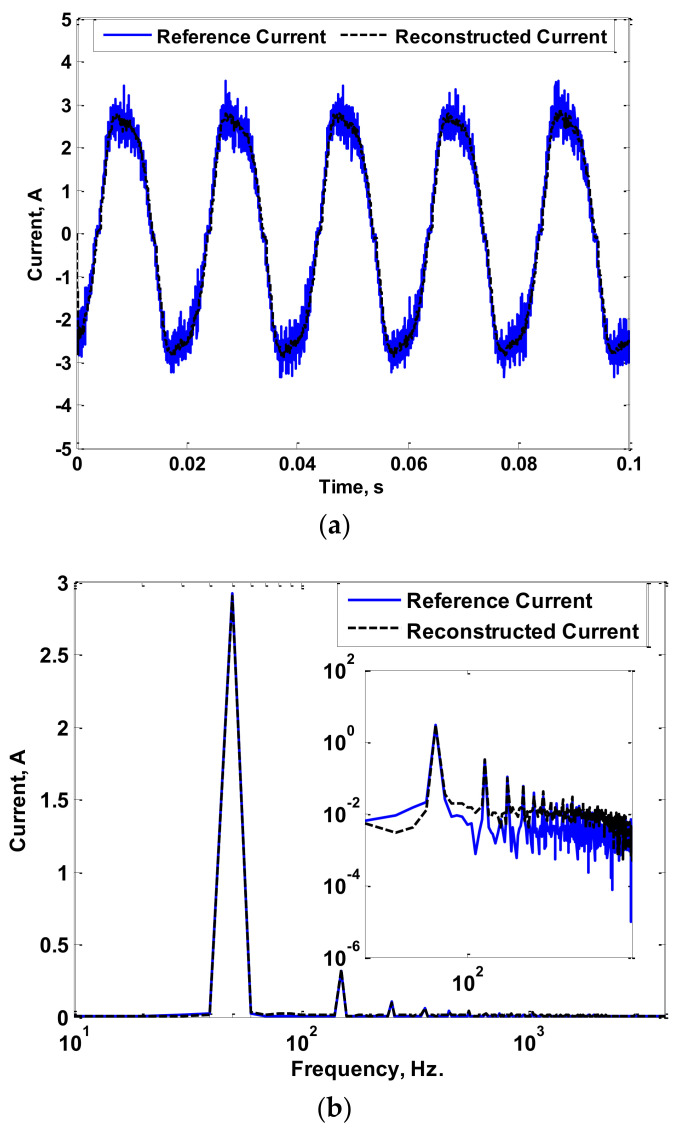
Reference and Reconstructed Currents of a 400 W luminaire: (**a**) Reference and Reconstructed Current Waveforms; (**b**) FFT of Reference and Reconstructed Currents.

**Figure 15 sensors-22-04220-f015:**
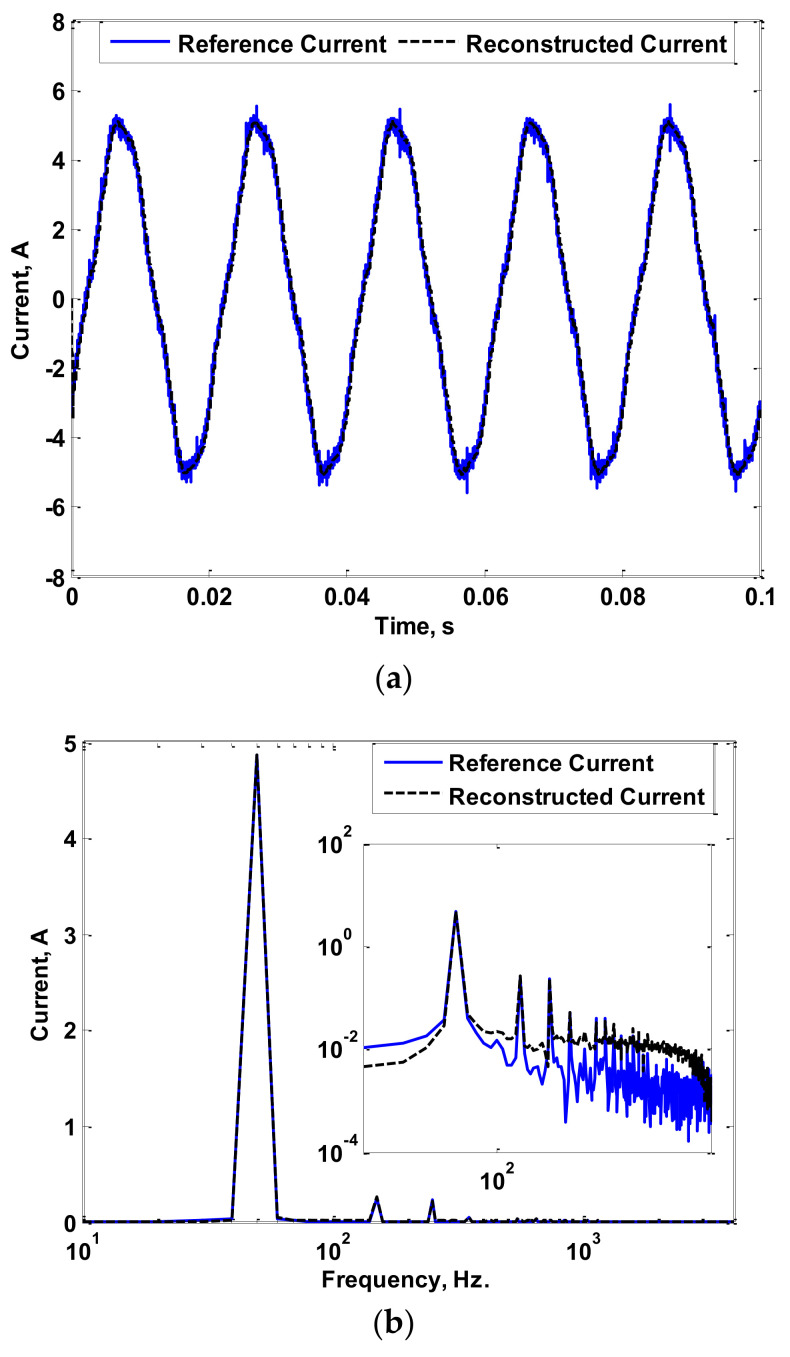
Reference and Reconstructed Currents of a 600 W luminaire: (**a**) Reference and Reconstructed Current Waveforms; (**b**) FFT of Reference and Reconstructed Currents.

**Figure 16 sensors-22-04220-f016:**
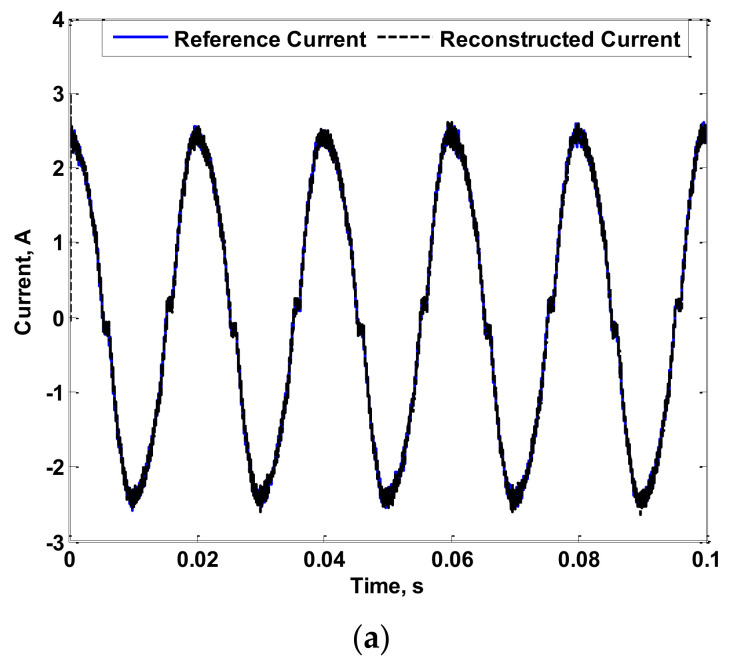
Reference and Reconstructed Currents of a 300 W luminaire Considering 35 kHz: (**a**) Reference and Reconstructed Current Waveforms; (**b**) FFT of Reference and Reconstructed Currents.

**Table 1 sensors-22-04220-t001:** Errors in Measurement in Fundamental and Harmonic Components for the 300 W, 400 W, and 600 W LED Luminaires.

Harmonic Order	% Error		
	300 W LED	400 W LED	600 W LED
Fundamental	0.42	0.34	0.184
3rd Harmonic	6.71	−2.66	−6.23
5th Harmonic	−3.72	7.9	4.53
7th harmonic	−3.43	−7.63	−2.56

## Data Availability

Not Applicable.

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
