# Peer review of "Measurement of Power Frequency Current including Low- and High-Order Harmonics Using a Rogowski Coil"

_sensors, 2022, doi:10.3390/s22114220_

Round 1

Reviewer 1 Report

1)The paper has not been formatted according to the MDPI standard

2)The paper deal with contactless current sensor using Rogowski Coil

3) A proper introduction  and comparison with the other contactless (with pro and contra) is missing. The author should briefly cite/comment/discuss this current paper as well as most of the papers cited by this paper (on Rdson, contactless and Hall-effect sensors)... ..and also add other relevant papers that they may find in the literature. In brief, a deep comparison and description of the state-of-the-art is missing.
Referring to contactless sensor, the author may refer to typical EMI issues and countermeasures (i.e. googling "EMI and current sensors": 
https://doi.org/10.1016/j.microrel.2012.10.013  and others from MDPI). 

4) The paper lacks a deep comparison with the state of the art, also regarding the same Rogowski coil. Accordingly, please drastically increase the reference list that is now very poor. 

5) The more paper are cited regarding the same topic,  the better ( notice that MDPI has no pages limit). For a rough search on the Electronic MDPI papers, a similar and recent paper on current detection on power-electronics circuits is 10.3390/electronics11010009 and 10.3390/electronics10172143

6) The work is somehow interesting as it is based on measurements, but please highlight better which is the scientific novelty in your work (if any).  It looks like a project/experiment dissertation more than a scientific paper. 

Author Response

The authors are grateful to the editor and reviewers for giving us the opportunity to improve our manuscript. Based on the concerns and suggestions, modifications are carried out in a red colour the revised manuscript. Our response to reviewers according to the comments is presented as follows:

Point 1: The paper has not been formatted according to the MDPI standard.

Response 1: The revised version is formatted according to the MDPI standard.

Point 2: The paper deal with contactless current sensor using Rogowski Coil.

Response 2: Ok, I agree with you.

Point 3: A proper introduction  and comparison with the other contactless (with pro and contra) is missing. The author should briefly cite/comment/discuss this current paper as well as most of the papers cited by this paper (on Rdson, contactless and Hall-effect sensors)... ..and also add other relevant papers that they may find in the literature. In brief, a deep comparison and description of the state-of-the-art is missing.

Referring to contactless sensor, the author may refer to typical EMI issues and countermeasures (i.e. googling "EMI and current sensors":

https://doi.org/10.1016/j.microrel.2012.10.013  and others from MDPI).

Response 3: New paragraphs are added to the introduction section to compare with other contactless sensors. This results in adding new references. The new number of references become 33, however, the old references are 23 .

Point 4: The paper lacks a deep comparison with the state of the art, also regarding the same Rogowski coil. Accordingly, please drastically increase the reference list that is now very poor.

Response 4: Brief comparison with other type of sensors such as current transformers and Hall effect sensor. Also, comparison between reconstruction methods are added. This results in adding new references. The new number of references become 33, however, the old references are 23 .

Point 5: The more paper are cited regarding the same topic,  the better ( notice that MDPI has no pages limit). For a rough search on the Electronic MDPI papers, a similar and recent paper on current detection on power-electronics circuits is 10.3390/electronics11010009 and 10.3390/electronics10172143.

Response 5: New references considering related papers from different journals and MDPI journals are added.

Point 6: The work is somehow interesting as it is based on measurements, but please highlight better which is the scientific novelty in your work (if any).  It looks like a project/experiment dissertation more than a scientific paper.

Response 6: Great thanks for you. The novelty of the paper is more highlighted in the introduction part. Generally, the contribution of the paper can be summarized in the following paragraph:

"In fact, there are several methods used for current reconstruction [4, 10, 30-32]. In [4], evaluation of digital integration is carried out. In the same research [4], digital Fourier transform (DFT) is presented to extract the fundamental component of power frequency AC current. Also, the differential current reconstruction technique presented in  [5] is suitable only for measuring the fundamental component of the power frequency AC current that is intended to be measured. Up till now, using Rogowski coil to measure power frequency AC current including low and high order harmonics isn't carried out. Therefore, there is a need for a current reconstruction technique which has the ability to measure power frequency AC current containing low and high order harmonic components. Also, the published current reconstruction techniques depend on the measured coil parameters which can cause errors in measurement especially with the possibility of parameters variation with temperature and frequency. So, in this paper, measurement of power frequency AC current including low and high order harmonics is presented. The measurement is carried out using a Rogowski coil due to its advantages. The current of LED luminaire is chosen as it contains low and high order harmonics. The measurement process is dependent on a reconstruction technique that uses the measured characteristics of the Rogowski coil itself instead of the dependence on the coil parameters as it can be affected by frequency".

The references in the above paragraph are listed in the main manuscript with the same numbers.

Reviewer 2 Report

his paper presents the measurement of power frequency current of light emitting diode (LED) luminaires. Theoretical evaluation and experimental evalution are studied. The results are useful. However, several limitations need to be overcome.
1.  Why choose the power ratings of 300 w, 400 w and 600 w.
2.  Why the %Error decreases with the increase of power.
Theoretical evaluation is carried out using MATALB SIMULINK software. Explain in details.

Author Response

The authors are grateful to the editor and reviewers for giving us the opportunity to improve our manuscript. Based on the concerns and suggestions, modifications are carried out in a red colour the revised manuscript. Our response to reviewers according to the comments is presented as follows:

Point 1: Why choose the power ratings of 300 w, 400 w and 600 w.

Response 1: In fact, there are several power ratings of LED luminaires. The power ratings 300 W, 400 W and 600 W are widely used in street and stadiums. Also, they are chosen as to have currents more than 1 A as the experimental Rogowski coil is designed to measure currents equal or more than 1 A accurately.

Point 2: Why the %Error decreases with the increase of power.

Response 2: The % Error decreases with the increase of power. Hence, the increase in power results in an increased current. The increase in current results in an increased magnetic field which results in a greater flux linkage with the Rogowski coil winding. Therefore, a better sensitivity and accuracy is achieved.

Point 3: Theoretical evaluation is carried out using MATALB SIMULINK software. Explain in details.

Response 3: The Rogowski coil is modeled using MATLAB SIMULINK adopting its equivalent circuit. This point is more clarified in the revised version. So, a new figure (Figure 4) is added in the revised version.

Round 2

Reviewer 1 Report

The paper has been improved. Anyhow, the authors neglected the suggestion to include in the introduction and overview of the alternative method that can be used instead of the Rogowski coil. In other term: why and in which case the use of the Rogowski coil is better? what are the pro and contra respect to other methods (to be cited). Also, the suggested references have been neglected.

The number of references for a more fair comparison table should be further increased. Please consider in particular Sensor MDPI paper on the same topic (current sensing)
